# Validation of a novel handheld lactate analyzer reveals concentration-dependent bias compared with a laboratory reference device

Benedikt Meixner[1,2,3,*], Manuel Matzka[4], Peter Renner[3], Silvia Achtzehn[5] and Billy Sperlich[1]

## ABSTRACT

This study aimed to validate a handheld capillary blood lactate analyzer (Lactate Express) against a reference (EKF Biosen C-Line) and assess its impact on lactate thresholds. Thirty-five participants completed incremental running or cycling tests with capillary sampling. Lactate was measured with the Lactate Express and an EKF Biosen C-Line analyzer, yielding 231 paired samples (0.5–14.2 mmol l$^{-1}$). Agreement was evaluated using intraclass correlation, Lin's concordance correlation, mean absolute error, and Bland-Altman analyses overall and within <1, 1–4, and >4 mmol l$^{-1}$. Thresholds at 2 and 4 mmol l$^{-1}$ were compared between devices for 18 participants in both exercise modes. Overall agreement was high (intraclass correlation coefficient 0.98), but Bland-Altman analysis showed a bias of −0.29 mmol l$^{-1}$ with concentration dependent behavior. Bias was negligible within the 1–4 mmol l$^{-1}$ range (−0.05 mmol l$^{-1}$), positive at concentrations <1 mmol l$^{-1}$ (0.29 mmol l$^{-1}$) with pronounced underestimation at concentrations >4 mmol l$^{-1}$ (−1.03 mmol l$^{-1}$). Differences in 2 and 4 mmol l$^{-1}$ thresholds were small and non-systematic. The Lactate Express demonstrates good relative and absolute agreement within the 1–4 mmol l$^{-1}$ range relevant for threshold-based training prescription, but exhibits substantial and systematic bias at both low and high lactate concentrations. The device may provide broadly comparable threshold estimates in some cases but should not be considered interchangeable with the laboratory reference device across the full lactate concentration spectrum.

KEY WORDS: Lactate, Handheld device, Validation

## INTRODUCTION

Capillary blood lactate (capBLa) is widely used in both clinical and performance settings as an indicator of metabolic stress, tissue hypoxia, and exercise intensity (Swart and Jennings, 2004). The practical application of lactate measurements hinges on multiple factors, including biological variability (Bagger et al., 2003), sampling site consistency (Raa et al., 2020), protocol standardization (Bourdon et al., 2018), and – fundamentally – the reliability and validity of the capBLa analyzers themselves. While stationary laboratory-grade devices are widely used as reference methods (Mentzoni et al., 2024; Davison et al., 2000; Hart et al., 2013; Zhong et al., 2025), the demand for portable, field-deployable solutions has increased as lactate-guided intensity regulation has become more integrated into training practice.

Handheld analyzers for capBLa offer practical advantages for real-time, on-site testing, yet their accuracy and precision remain subjects of debate. Prior studies report conflicting findings, with some highlighting limitations in precision compared to stationary systems (Swart and Jennings, 2004; Bonaventura et al., 2015); meanwhile, others demonstrate comparable performance (Tanner et al., 2010; Crotty et al., 2021). These discrepancies underscore the need for rigorous validation of emerging handheld technologies to ensure their suitability for athletic and health monitoring.

In this study, we evaluated the analytical performance of a novel handheld lactate analyzer (Lactate Express) by determining its accuracy and precision relative to an established stationary reference method (EKF Biosen C-Line). According to the manufacturer, the handheld device simultaneously measures blood lactate and glucose concentrations using only 0.8 µl of capillary blood. Lactate determination is based on an electrochemical biosensor technology (Nikolaus and Strehlitz, 2008) with a reported measurement range of 0.5–28 mmol l$^{-1}$.

The aim of this article was to validate the novel handheld Lactate Express by comparing its analytical agreement, precision, and accuracy of capBLa against the conventional, laboratory-based EKF Biosen C-Line as point-of-care reference device and to assess the implications for lactate threshold-based exercise testing.

[1]Julius-Maximilians-Universität Würzburg, Integrative and Experimental Exercise Science and Training, Department of Sport Science, Judenbühlweg 11, 97082 Würzburg, Germany. [2]Friedrich-Alexander-Universität Erlangen-Nürnberg, Department for Sport Science and Sport, Gebbertstraße 123b, 91058 Erlangen, Germany. [3]iq-move Praxis Dr. Fraunberger, Gebbertstraße 123b, 91058 Erlangen, Germany. [4]Department of Management and Sport, IST-University of Applied Sciences, 40233 Dusseldorf, Germany. [5]Schwarz Corporate Solutions KG, Health and Performance, Stiftsbergstraße 1, 74172 Neckarsulm, Germany.

*Author for correspondence (benedikt.meixner@fau.de)

B.M., 0000-0001-7044-9426; M.M., 0000-0002-2771-3030; B.S., 0000-0003-4686-8561

## RESULTS

Descriptive results are presented in Table 1, and all measured data points are presented in Fig. 1. Intraclass correlation coefficient (ICC) was 0.98 (95% CI: 0.96–0.98), and concordance correlation coefficient (CCC) was 0.95, while mean absolute error (MAE) was calculated as 0.55 (95% CI: 0.46–0.65), indicating that the typical absolute between-device deviation was approximately 0.5 mmol l$^{-1}$ across the full dataset.

Across all paired samples, Bland–Altman analysis showed a mean bias of −0.29 mmol l$^{-1}$ with limits of agreement (LoA) from −1.97 to 1.38 mmol l$^{-1}$.

Bland–Altman analysis determined a bias of 0.29 mmol l$^{-1}$ and LoA of −0.05 to 0.63 mmol l$^{-1}$ for the range from 0.5 to 1 mmol l$^{-1}$, a bias of −0.05 mmol l$^{-1}$ and LoA of −0.72 to 0.62 mmol l$^{-1}$ for the range from 1 to 4 mmol l$^{-1}$ and a bias of

**Table 1. Descriptive lactate values for EKF Biosen C-Line and Lactate Express**

| | EKF Biosen C-Line | Lactate Express |
|---|---|---|
| $n$ | 231 | 231 |
| Median (mmol l$^{-1}$) | 2.11 | 2 |
| Interquartile range (mmol l$^{-1}$) | 3.62 | 3.1 |
| Minimum (mmol l$^{-1}$) | 0.5 | 0.7 |
| Maximum (mmol l$^{-1}$) | 14.2 | 12.3 |

**Table 2. Results for Bland–Altman analysis with bias and limits of agreement for all lactate values and ranges below 1, 1–4 and above 4 mmol l$^{-1}$ measured on the EKF Biosen C-Line device**

| | Bias (mmol l$^{-1}$) | Lower LoA (mmol l$^{-1}$) | Upper LoA (mmol l$^{-1}$) | $n$ |
|---|---|---|---|---|
| All | −0.29 | −1.97 | 1.38 | 231 |
| <1 (mmol l$^{-1}$) | 0.29 | −0.05 | 0.63 | 36 |
| 1–4 [mmol l$^{-1}$] | −0.05 | −0.72 | 0.62 | 125 |
| >4 [mmol l$^{-1}$] | −1.03 | −3.33 | 1.28 | 70 |

−1.03 mmol l$^{-1}$ and LoA of −3.33 to 1.28 mmol l$^{-1}$ for the range above 4 mmol l$^{-1}$. Bias and LoA for all values, as well as analyzed ranges, are displayed in Table 2. Bland–Altman plots are displayed for all values in Fig. 2 and separately for analyzed ranges in Fig. 3A–C. All collected paired samples are included in Dataset 1; 95% CIs of Bland–Altman analysis are included in Table S1.

Differences in 2 and 4 mmol l$^{-1}$ threshold calculations in a subset of 18 participants are displayed in Fig. 4 for running ($n$=9, 2 female; panel A) and cycling ($n$=9, 4 female; panel B).

## DISCUSSION

The aim of this study was to compare a novel handheld lactate analyzer with the established EKF Biosen C-Line reference device. The results revealed substantial LoA and a divergent bias across lactate concentration ranges. To evaluate performance across physiologically relevant domains, the analysis was divided into three ranges, using the EKF Biosen C-Line as the reference method.

For values below 1 mmol l$^{-1}$, the Lactate Express showed a systematic positive bias of 0.29 mmol l$^{-1}$, indicating a slight overestimation of low capBLa concentrations, with comparatively narrow LoA of 0.05–0.63 mmol l$^{-1}$. Within the range 1–4 mmol l$^{-1}$, which typically corresponds to the determination of lactate thresholds, no significant bias (−0.05 mmol l$^{-1}$) was observed; however, the LoA (−0.72 to 0.62 mmol l$^{-1}$) were wider than those reported for comparable handheld analyzers in previous studies (Bonaventura et al., 2015; Mentzoni et al., 2024). For values above 4 mmol l$^{-1}$, representing high-intensity exercise and substantial lactate accumulation, the Lactate Express demonstrated a negative bias of −1.03 mmol l$^{-1}$, reflecting an underestimation of blood lactate relative to the reference device, accompanied by large LoA (−3.33 to 1.28 mmol l$^{-1}$).

In a study comparing four handheld lactate devices (Lactate Plus, Lactate Pro2, Lactate Scout 4 and TaiDoc TD-4289) to an EKF

Biosen device (Mentzoni et al., 2024), the handheld systems appeared to demonstrate narrower intervals than those observed in the present study. It should be noted that prediction intervals and Bland–Altman LoA describe different statistical quantities and are not directly comparable. As such, differences in interval width should be interpreted with caution and not taken as evidence of superior agreement. In addition, the lactate concentration range assessed by Mentzoni et al. (2024) (0.88–4.89 mmol l$^{-1}$) was relatively restricted. Given the present findings of concentration-dependent bias, with greater discrepancies observed at lower and higher concentrations, this narrower ranges may reduce observed variability, as it does not capture the greater discrepancies that may occur across a wider physiological range.

In a comparison of handheld devices to a blood gas analyzer (Radiometer ABL90) as criterion, six handheld devices (Lactate Pro, Lactate Pro2, Lactate Scout+, Xpress™, Edge, and i-Stat) consistently showed narrower LoA across all ranges of lactate concentrations (Bonaventura et al., 2015). The Lactate Pro2 was also validated against a YSI 1500 Sport device and displayed narrower LoA across ranges of <4, 4–8 and >8 mmol l$^{-1}$ (Crotty et al., 2021).

Based on our data and the available literature, we conclude that the novel handheld analyzer demonstrates acceptable agreement with the laboratory reference only within a limited range of physiological lactate concentrations. While the systematic overestimation below 1 mmol l$^{-1}$ and underestimation above 4 mmol l$^{-1}$ resemble patterns seen in other handheld devices, the magnitude of the bias and the width of the LoA in our study indicate reduced precision, particularly at higher concentrations where accurate quantification is most relevant for determining training intensity or metabolic thresholds.

In contrast to Bland–Altman analysis, which informs on systematic bias, LoA, and concentration-dependent disagreement, the MAE provides a simple estimate of the typical absolute mismatch between analyzers in the original unit. In the present study, the MAE was 0.55 mmol l$^{-1}$ (bootstrap 95% CI: 0.47–0.65 mmol l$^{-1}$), indicating

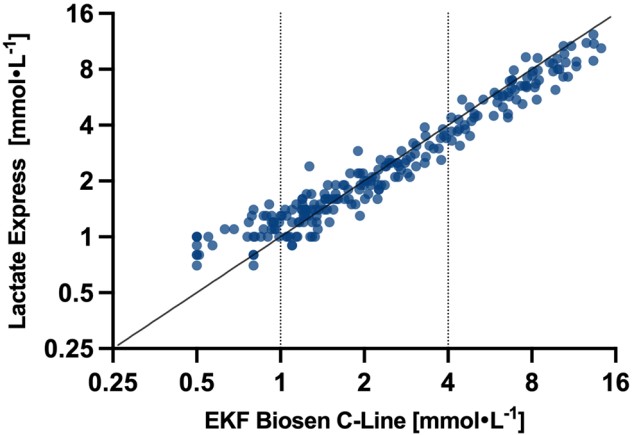

**Fig. 1. Scatter plot of all measured data points for EKF Biosen C-Line compared to Lactate Express, displayed on a logarithmic scale.**

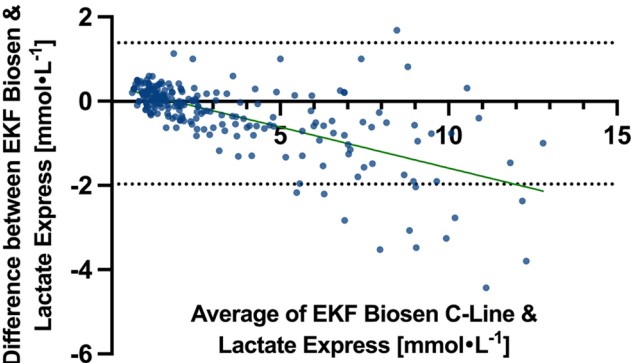

**Fig. 2. Bland–Altman plot with bias and limits of agreement, including 95% CIs for all capillary blood lactate values.**

Biology Open

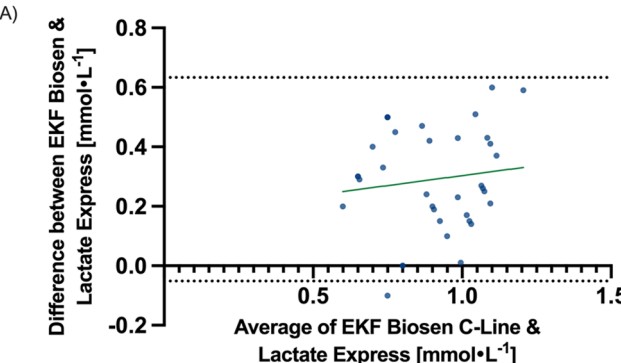

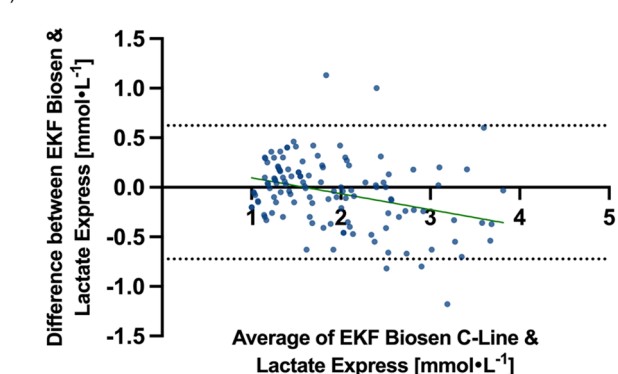

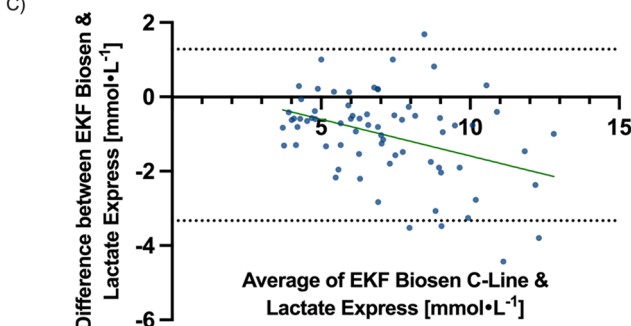

**Fig. 3. Bland–Altman plot with bias and limits of agreement, including 95% CIs for lactate values measured on the EKF Biosen C-Line device.** (A–C) Lactate values below 1 mmol l⁻¹ (A), between 1 and 4 mmol l⁻¹ (B) and above 4 mmol l⁻¹ (C).

that individual readings differed by approximately 0.5 mmol l⁻¹ on average. This may be particularly useful from a practical perspective, as it reflects the approximate magnitude of deviation a user may expect for an individual lactate reading. However, unlike Bland–Altman analysis, the MAE does not indicate the direction of disagreement and is therefore complementary rather than interchangeable.

Importantly, the high ICC and CCC observed in the present study should be interpreted alongside, rather than instead of, the Bland–Altman results. In our dataset, these coefficients indicated excellent overall consistency and concordance between the Lactate Express and the EKF Biosen C-Line across the full range of paired lactate values, suggesting that the handheld device generally preserved the ranking and overall pattern of measurements. However, this does not imply analytical interchangeability. In contrast, Bland–Altman analysis provides information on absolute agreement, revealing

systematic bias, the range of between-device differences, and their concentration dependence. Accordingly, high ICC and CCC values can co-exist with practically relevant disagreement when a device measures consistently but deviates in absolute terms, particularly at lower and higher lactate concentrations.

From a practical standpoint, this discrepancy in absolute agreement implies that the device can be used for screening or educational purposes or for trend monitoring within individuals, but not interchangeably with laboratory instruments when accurate absolute lactate values are required. Users should therefore interpret single values with caution, prioritize consistency in the choice of analyzer within a testing protocol, and avoid mixing handheld and laboratory data when establishing or monitoring lactate thresholds. In our data, when employing fixed lactate thresholds of 2 and 4 mmol l⁻¹, we found substantial differences, of e.g. ∼15 W in cycling or ∼1.5 km h⁻¹ in some cases, while other lactate curves revealed close to no difference (Fig. 4).

It should be noted that data availability differed across concentration ranges, with fewer observations at the lower and upper ends of the lactate spectrum than in the 1–4 mmol l⁻¹ range. This likely reduced the precision of the range-specific bias and LoA estimates, particularly at the concentration extremes. However, because bias remained close to zero in the well-populated 1–4 mmol l⁻¹ range and was clearly shifted in opposite directions below 1 and above 4 mmol l⁻¹, unequal data availability alone is unlikely to explain the observed concentration-dependent disagreement.

The present study should also be interpreted considering its application focus. Although lactate assessment has utility beyond threshold-based exercise testing, including broader bioenergetic characterization, monitoring of metabolic strain, field-based athlete monitoring, and clinical or point-of-care settings, we specifically evaluated device performance in the context of incremental exercise testing and fixed lactate reference points of 2 and 4 mmol l⁻¹. These values were chosen because they remain widely used pragmatic anchors for exercise-intensity prescription, but they do not encompass all contexts in which mobile lactate analyzers may be applied. Accordingly, the present findings are most directly relevant to threshold-based testing, whereas the suitability of the device for other applications, including bioenergetic profiling or repeated monitoring in other settings, requires further study. Furthermore, our method is limited to testing at the earlobe; other sampling sites such as the finger might produce different results. Additionally, the threshold comparison should be treated with caution as only 18 participants were included in this analysis.

One limitation is that the dataset included repeated paired measurements from the same participants, such that observations were not fully independent. This within-subject dependency was not modeled explicitly and may have influenced the precision of some agreement estimates, particularly the ICC and the range-specific Bland–Altman LoA. In addition, the concentration ranges were defined pragmatically to reflect low, moderate, and high lactate values for practical interpretation, rather than being based on formal physiological or statistical cut points.

The larger bias at lower and higher lactate concentrations may reflect a combination of analytical and pre-analytical factors. Previous studies (Bonaventura et al., 2015; Tolan et al., 2017; Tanner et al., 2010) have shown that handheld lactate analyzers commonly become increasingly negatively biased at higher lactate concentrations, which is consistent with concentration-dependent limitations of strip-based electrochemical measurement or calibration. At the lower end of the range, small absolute deviations and capillary sampling effects may become proportionally more influential.

Biology Open

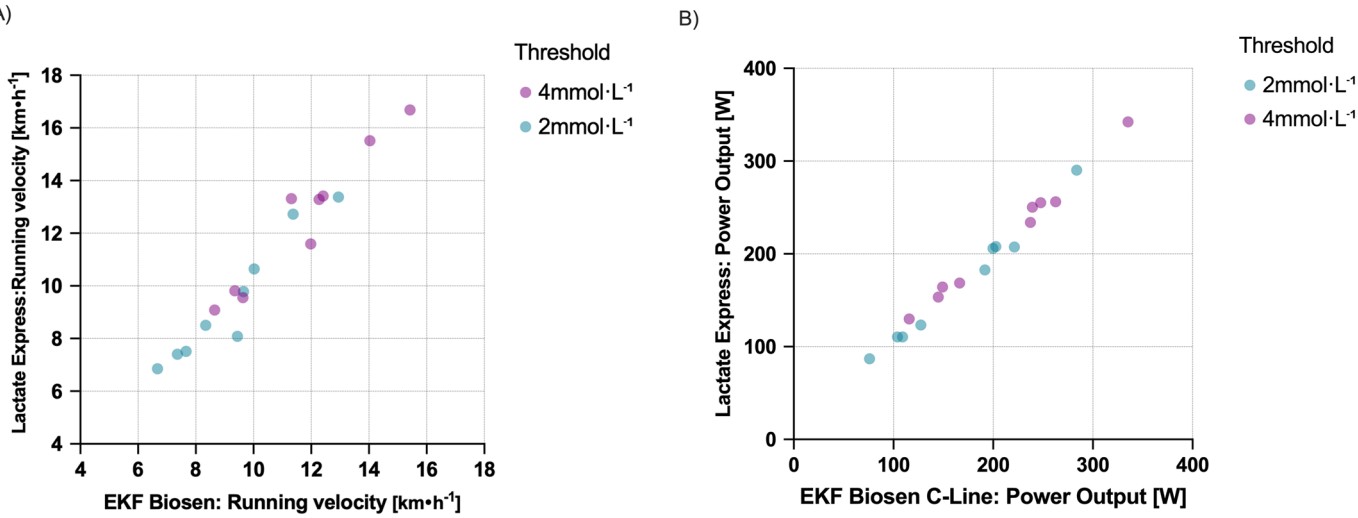

**Fig. 4. Scatter plots of running velocity and cycling power output at onset of 2 and 4 mmol l⁻¹ capillary blood lactate concentration determined by the Lactate Express and EKF Biosen C-Line devices.** (A) Running velocity. (B) Cycling power output.

In addition, handheld systems are more susceptible to very small sample volumes, incomplete strip filling, and capillary-sampling artefacts than bench-top analyzers. These issues should be explored directly in future validation studies.

A further limitation is that the investigated device was not commercially purchased but provided by the distributor in connection with a symposium presentation by one of the authors. The study was initiated out of practical curiosity regarding the device's potential usefulness for field-based testing. While the manufacturer and distributor had no involvement in study design, data collection, analysis, interpretation, or manuscript preparation, this circumstance may still represent a perceived conflict of interest.

Based on the present findings and study protocol, the Lactate Express demonstrated acceptable relative agreement but only limited absolute agreement with the laboratory reference device within a limited range of physiological lactate concentrations. While the analyzer showed smaller disagreement at low to moderate levels, systematic overestimation below 1 mmol l⁻¹ and underestimation above 4 mmol l⁻¹ indicate a concentration-dependent bias. The wide LoA at higher lactate concentrations further suggest reduced precision in conditions of high metabolic stress. While disagreement was smaller at low to moderate concentrations, systematic overestimation below 1 mmol l⁻¹ and underestimation above 4 mmol l⁻¹ indicate a concentration-dependent bias. Consequently, the device may be of interest for selected field-based applications, but it should not be considered interchangeable with the laboratory reference device across the full range of lactate concentrations.

## MATERIALS AND METHODS
A convenience sample of 35 men (n=27) and women (n=8) participated in this study. All subjects completed incremental exercise tests in the laboratory, performed for either health assessment or performance diagnostics. Participants underwent further health-related diagnostic procedures not related to this study and would have been excluded if any indications of conditions potentially influencing lactate metabolism or kinetics were found, but this was not the case.

The incremental protocol elicited a progressive increase in exercise intensity, thereby inducing a wide physiological range of capBLa concentrations from low to high levels. Laboratory conditions were maintained between 20.0°C and 22.0°C, with relative humidity ranging from 20% to 45% across all testing days. Written informed consent was obtained from all adult participants, and all procedures were approved by the

Ethical Committee of Exercise Science and Training of the Faculty of the University of Würzburg (EV2025/6-1611) in accordance with the Declaration of Helsinki.

### Participants
Participants were recruited from routine clinical practice according to their health status and the availability of the laboratory assistant. Participant characteristics are presented in Table 3.

### Testing protocol
All participants performed either a running (n=18) or cycling (n=17) incremental step test with 3-min stages. Based on individual fitness level, participants began at 25, 50, or 75 W with 25 W increments for cycling on a Cyclus2 (RBM elektronik-automation GmbH, Leipzig, Germany) ergometer or at 4, 6, or 8 km h⁻¹ with 2 km h⁻¹ increments for running on a treadmill (quasar, h/p/cosmos, Nußdorf-Traunstein, Germany). Running protocols incorporated a 30-s break in between stages to allow for lactate sampling. This incremental design ensured the inclusion of workloads eliciting lactate concentrations from resting to high-intensity exercise levels. All protocols were selected because of their habitual use in sports medicine practice and their adoption in previous studies (Roecker et al., 2003; Dantas and Doria, 2015; Kantanista et al., 2016; Madrid et al., 2016; Messonnier et al., 2013).

### capBLa measurements
A total of n=231 capillary blood samples collected at the end of each 3-min stage during the incremental tests were analyzed. At the end of every stage, the earlobe was wiped clean, and any residual blood was removed; after, as few drops of capillary blood as possible were obtained from the left earlobe, enabling measurements using both devices: 0.8 µl of blood for the Lactate Express (Eaglenos, Nanjing, China) and 20 µl for the EKF Biosen C-Line (EKF Diagnostics, Barleben, Germany) analyzer. To minimize the risk of systematic measurement bias, we alternated the order of capillary blood samples for every measurement. Participants were not considered for this

**Table 3. Participants' characteristics as mean±s.d.**

| Variable | All | Included in threshold analysis (n=18) |
|---|---|---|
| VO₂ peak (ml/kg/min) | 46.5±9.1 | 47.7±9.6 |
| Body mass (kg) | 77.5±12.9 | 76.1±12.0 |
| Age (years) | 37.4±17.8 | 34.6±16.5 |
| Stature (cm) | 177±7 | 177±7 |
| Body fat (%) | 18.8±8.1 | 16.8±8.1 |

study if it became evident that blood sampling would prove to be challenging in a timely manner. The Lactate Express was calibrated and verified against high (14.5–18.9 mmol l⁻¹) and low (3.5–5.9 mmol l⁻¹) reference solutions from the manufacturer, yielding calibration results within the provided range. The EKF Biosen C-Line was calibrated and controlled hourly using standard and control solutions according to the manufacturer's specifications.

## Lactate threshold calculations

For the determination of lactate thresholds, capBLa was plotted against cycling power or running velocity. Fixed lactate concentrations of 2 and 4 mmol l⁻¹ were selected as widely used pragmatic reference points in exercise testing and training prescription, allowing us to examine the practical consequences of between-device measurement differences. A third-order polynomial regression was performed, and the power output or running velocity at lactate concentrations of 2 and 4 mmol l⁻¹ was determined (Heuberger et al., 2018). This analysis was performed only when an acceptable number of measurements (≥6) was available, and no values were missing. Due to the fact that this data collection was performed during regular medical cardiopulmonary exercise testing, it was not always possible to ensure no missing values during data collection for every single patient. Nine runners and nine cyclists were analyzed.

## Statistical analysis

Statistical analysis was performed for three ranges of capBLa measured using the EKF Biosen C-Line: 0.5–1 mmol l⁻¹ ($n$=36), 1–4 mmol l⁻¹ ($n$=125) and >4 mmol l⁻¹ ($n$=70).

The ICC was calculated following the recommendations of Koo and Li (2016), as an index of between-device reliability, treating the two lactate analyzers as two raters measuring the same samples.

We used a two-way random-effects, single-measurement model for absolute agreement, such that the estimate reflects both random error and any systematic differences between devices. The CCC was determined according to Lin (1989) to quantify both precision and accuracy relative to the line of identity. Bland–Altman analysis was performed according to Bland and Altman (1986) to assess between-device agreement, including computation of the mean bias and 95% LoA (mean±1.96 s.d.). The MAE was calculated following Willmott and Matsuura (2005), representing the average magnitude of absolute differences between paired measurements.

## Competing interests

The Lactate Express device evaluated in this study was provided by the distributor in connection with a symposium presentation by one of the authors (P.R.), and the study was initiated out of practical curiosity regarding the device's potential usefulness for field-based testing. Neither the manufacturer, the distributor, nor any of their employees had any role in the study design, data collection, data analysis, interpretation of the results, manuscript preparation, or the decision to submit the manuscript. P.R. and B.M. are affiliated with iq-move Praxis Fraunberger, where the study was conducted. S.A. contributed expertise in laboratory devices and diagnostics as a co-author and is affiliated with Schwarz Corporate Solutions K.G., Health and Performance. These affiliations did not influence the collection, analysis, or interpretation of the data.

## Author contributions

Conceptualization: B.M., B.S.; Data curation: B.M.; Investigation: B.M., M.M., B.S.; Methodology: P.R., B.S.; Resources: P.R., B.S.; Validation: B.M., M.M., P.R., S.A.; Visualization: B.M.; Writing – original draft: B.M.; Writing – review & editing: M.M., S.A., B.S.

## Funding

 Deposited in PMC for immediate release.

## Data and resource availability

All relevant data and details of resources can be found within the article and its supplementary information.

## Peer review history

The peer review history is available online at https://journals.biologists.com/bio/lookup/doi/10.1242/bio.062543.reviewer-comments.pdf

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
