## [Peer Review File · Biology Open]

Validation of a Novel Handheld Lactate Analyzer Reveals Concentration-Dependent Bias Compared with a Laboratory Reference Device

Manuel Matzka, Peter Renner, Silvia Achtzehn, Billy Sperlich and Benedikt Johannes Meixner

DOI: 10.1242/bio.062543

Editor: Kendra J. Greenlee

Review timeline

Original submission:	25 February 2026
Editorial decision:	6 March 2026
First revision received:	16 March 2026
Editorial decision:	22 March 2026
Second revision received:	23 March 2026
Accepted:	24 March 2026

Original submission

First decision letter

MS ID#: bio.062543

MS Title: Validation of a Novel Handheld Lactate Analyzer Reveals Concentration-Dependent Bias Compared with a Laboratory Reference Device

Authors: Benedikt Johannes Meixner, Manuel Matzka, Peter Renner, Silvia Achtzehn, Billy Sperlich

I have now reached a decision on the above manuscript. The reviewer reports are shown at the bottom of this email. As you will see, the reviewers raised a number of substantial criticisms that prevent me from accepting the paper at this stage.

They suggest, however, that a revised version might prove acceptable, if you can address their concerns. If you think that you can deal satisfactorily with the criticisms on revision, I would be pleased to see a revised manuscript.

In addition to the reviewers' comments, there is a concern about the conflict of interest. Transparency is important in conflicts of interest, whether real or perceived. In this case, the conflict of interest requires more prominent and critical acknowledgement throughout the paper. The Lactate Express was provided by the manufacturer's distributor in exchange for a symposium presentation by one of the authors, Dr. Renner. While this is disclosed at the end of the manuscript, its implications for the interpretation of results are not adequately discussed in the body of the manuscript. The overall framing of the device as suitable for threshold monitoring (despite having wider limits of agreement than comparable devices in the literature) should be presented with appropriate caution given this potential source of bias. Independent replication with commercially purchased devices would strengthen the evidence, and this lack of independent replication should be pointed out to readers. These points should be explicitly discussed in a "Limitations" section in the manuscript.

Furthermore, the COI statement will need modification. The affiliations of two authors with iq-move Praxis Fraunberger and one author with Schwarz Corporate Solutions KG, Health & Performance need descriptions, to alleviate perceived conflicts of interest.

At this stage, we also ask you to ensure your manuscript complies with our formatting guidelines. Provided you are able to fully address the referees' comments, we are positive about publication of your paper (we accept over 95% of revision submissions) and therefore hope you won't mind any extra work involved in reformatting your manuscript at this point.

Please upload both a 'clean' version of your Word file, along with a highlighted version clearly showing where you have made changes in the revised manuscript. Please avoid using 'Track changes' in Word files as these are lost in PDF conversion.

I should be grateful if you would also provide a point-by-point response detailing how you have dealt with the points raised by the reviewers in the 'Response to Reviewers' box. Please attend to all of the reviewers' comments. If you do not agree with any of their criticisms or suggestions please explain clearly why this is so.

Reviewer 1

Comments for the author

This study addresses a practically relevant question regarding the analytical validity of a novel handheld blood lactate analyzer (Lactate Express) relative to an established point-of-care reference device (EKF Biosen C-Line). The stratified Bland-Altman approach across three physiologically meaningful concentration ranges is a clear strength of the manuscript and the inclusion of a threshold-based practical analysis adds meaningful applied value. However, there are several methodological, interpretive and presentational concerns that should be addressed before the manuscript is suitable for publication. The quality of the written English is generally acceptable, though a number of minor grammatical and typographical issues are noted below.

Abstract:

- * The abstract is generally well written and conveys the key findings.
- * However, a full stop is missing at the end of the Main results section ("–1.03 mmol·L⁻¹Differences"); this should be corrected.

Introduction:

The introduction is concise and adequately motivates the study. The relevant literature on handheld lactate device validation is cited and the aim is clearly stated. A few points should be addressed:

- * There is a full stop immediately prior to the reference in the first sentence. This should be removed.
- * A reference would be helpful to support the statement "While stationary laboratory-grade devices are widely regarded as the gold standard".
- * The introduction frames the EKF Biosen C-Line as an "established stationary reference" device. However, little detail is provided regarding this device. It is critical to the validity of the manuscript that the EKF Biosen C-Line is indeed the gold standard reference, so I would ask that the authors justify this selection.
- * The final paragraph of the introduction effectively states the aim twice (once in general terms and again in specific terms). Consider condensing these into a single, clearly structured aim statement.

Methods:

- * The participant group was heavily biased towards males (27 men and 8 women). This is an ongoing problem in the physiology literature, so the authors should provide a justification for this bias or acknowledge it as a limitation. Secondly, given that physiological differences between sexes could plausibly influence lactate kinetics and the distribution of concentration ranges in the dataset, the authors should either acknowledge this as a limitation or confirm that such an imbalance did not affect the results.
- * Participants were "recruited from routine clinical practice according to their health status and the availability of the laboratory assistant." This description raises questions about selection bias and the representativeness of the sample. Were any health-related exclusion criteria applied? Were participants screened for conditions that might affect lactate metabolism? This should be clarified.
- * Table 1 might be improved by splitting by sex.
- * Testing protocol: The city name "Leipzig" in the Cycclus2 ergometer description appears to be a typographical error; it should read "Leipzig."
- * Testing protocol: There is another rogue full stop prior to a reference (line 55). Please check throughout.
- * The Lactate Express was calibrated using a "manufacturer-provided reference solution, yielding satisfactory calibration results." What were the calibration target values? How was "satisfactory" defined? Please provide the specific calibration outcomes or at minimum state the acceptable tolerance criteria used.
- * The lactate threshold analysis was restricted to participants with ≥ 6 complete measurements and no missing values, resulting in only 8 runners and 8 cyclists being analysed. This is a notably small subset of the full cohort and limits the generalisability of the threshold comparison. Could the authors please report the characteristics (e.g., age, fitness level, lactate range) of this subset relative to the full sample, clearly state the sex of these 8 runners and 8 cyclists and more explicitly acknowledge the limited statistical power of this analysis.

Statistical analysis:

- * The analytical approach (ICC, CCC, MAE, and stratified Bland-Altman) is well chosen and clearly described. The use of a two-way random-effects model for absolute agreement is appropriate given the study design.
- * The sample sizes across concentration ranges are considerably unequal: $n=36$ ($<1 \text{ mmol}\cdot\text{L}^{-1}$), $n=125$ ($1-4 \text{ mmol}\cdot\text{L}^{-1}$) and $n=70$ ($>4 \text{ mmol}\cdot\text{L}^{-1}$). This imbalance is not acknowledged or discussed and may be particularly relevant in the $<1 \text{ mmol}\cdot\text{L}^{-1}$ range (where the smaller sample may reduce the reliability of the LoA estimate). The authors might discuss the implications of these unequal group sizes for the interpretation of range-specific agreement.

Results:

- * The results are concise and the figures are clear. The Bland-Altman plots are well presented, and the stratified analysis effectively illustrates concentration-dependent behaviour.
- * Table 1 is described in the text as containing "descriptive results," but the table as presented contains only participant characteristics. The descriptive lactate data (medians, ranges etc.) are in Table 2. Please ensure the in-text references to tables are consistent with the tables as labelled.
- * Table 3 has 2 blank lines at the bottom - I'd suggest this should be cleaned up.
- * The text mentions Figures 6 and 7 for the threshold comparisons ("displayed in figure 6 for running and figure 7 for cycling"), but only Figure 4 is present in the manuscript. This appears to be a figure numbering inconsistency that should be resolved.

* The image quality of some of the figures is really quite poor. This should be enhanced.

Discussion:

The discussion engages well with the prior literature and the concentration-range framing provides a useful structure. However, several points may require attention:

- * It seems as though there should be a new paragraph and line break at line 9/10?
- * Lactate Express is sometimes in italics, sometimes not. Please correct this inconsistency.
- * There should be a space after the full stop on line 22. Please correct and check throughout.
- * The practical recommendations offered in the discussion (e.g., "prioritize consistency in the choice of analyzer", avoid mixing handheld and laboratory data") are sensible and appropriate. However, this paragraph refers to Figures 6 and 7, which don't appear to exist.
- * The limitations paragraph is brief and would benefit from expansion. Please add the limitations I mention earlier in my review.

In summary, this is a useful and practically motivated validation study that addresses a clear gap in the literature. The stratified Bland-Altman approach is particularly commendable. Addressing the points above would substantially strengthen the manuscript.

Reviewer 2

Comments for the author

The authors have conducted a validation study comparing a handheld lactate analyzer with a laboratory reference device. They reported good relative agreement, but concentration dependant bias in terms of absolute agreement, with greater discrepancies between analysis approaches apparent at both higher and lower lactate concentrations.

Overall, the article approaches an interesting and relevant topic that aligns well with the methods and techniques article description. Hand-held devices such as the lactate analyser under investigation here have large utility in practice, or for field-research, but validation against reference devices is important. In my opinion, the authors have conducted an interesting and useful study, and their conclusions are balanced and directly informed by their results. There were some aspects of the manuscript that I felt would benefit from further elaboration or clarification, however. Several points points were implied rather than directly stated, which can lead to misunderstanding and impact accurate interpretation. I have detailed the points that I believe warrant additional attention below.

Specific Comments:

1. The article largely focuses on the use of the Lactate Express in an exercise physiology context, however as identified by the authors in their opening sentence, lactate assessment has utility in wide-ranging contexts, including, but not limited to exercise physiology. Considering that this is a general Biology journal, I believe that it should not be assumed that all readers will be familiar with concepts such as threshold identification and their implications for program design and that some elaboration on these concepts, and justification for the design decisions made (e.g., the relevance of the 2 and 4 mmol thresholds) as well as consideration of the other contexts in which these mobile lactate analysers may be used would be useful.
2. Methods section: There were some parts of the first two paragraphs that I was unclear on. Were the participants in this study purposefully recruited or was this a convenience sample conducted on individuals who were undertaking incremental tests for other purposes? The statement on lines 6 - 7

("performed either for health assessment or performance diagnostics") implies the latter, but it is important to be explicit on this point. Were any eligibility criteria applied to participant selection, e.g., in terms of age, health or training status? The statement on line 25 - 26 implies that there were ("recruited according to their health status") but the details are unclear. Greater precision, clarity and accuracy is required throughout the methods to ensure the reader fully understands the details of the study.

3. Methods - Capillary Blood Lactate measurements: It is stated that a single drop of capillary blood was obtained from the left earlobe, enabling simultaneous measurements using both devices. Considering the large difference in required volume (0.8 vs 20 μ L) how was volume separation managed in practice? Did you have any issues in ensuring sufficient volume for each analysis and were there any issues in time delay between sampling and measurement? The alternation of assessment order is likely to at least partly overcome any potential issues, however greater clarity on how the design was applied in practice would be useful.

4. Methods - Lactate threshold calculation section: If I have understood correctly the threshold analysis was conducted on 16 individuals, presumably those who were undergoing performance based tests and for whom a larger number of data-points were available. This is important information that warrants further consider, e.g., in the abstract it is implied that the threshold data is available for the full dataset of 35 participants. Conducting this analysis only on a substantially reduced participant subset has important statistical implications that should be considered.

5. Following on from the above point, I believe that data availability for the different assessments made is an important consideration that warrants some further attention. How much of an influence does sample size have on these analyses? It is interesting that the concentration ranges that showed the greater biases also had substantially fewer data-points for evaluation. Is it possible that data availability may have impacted these results?

6. The dataset includes 231 paired lactate measurements obtained from 35 participants, meaning that multiple observations originate from the same individuals and are likely to be highly correlated. From what I understand, the statistical analyses treat these observations as independent, which may potentially impact the agreement metrics, particularly the ICC and Bland-Altman limits of agreement. Was dependency considered within the analysis? If not, it would be useful to consider the potential influence of this on the main outcomes of interest.

7. What was the purpose of the MAE assessment, considering that the Bland-Altman provides similar information? I recognize that these analyses can provide related and complementary information, and it would be useful to be more explicit on their unique contributions to the main question. It would also be useful to calculate confidence intervals around the MAE to give an indication of certainty in the estimate and to elaborate on the implications of this analysis in the results and discussion.

8. Were any predefined acceptance criteria for the various analyses used? Based on the discussion, it appears not, and at several points somewhat vague phrases are used, e.g., "substantial limits of agreement" and LoA which were "somewhat wider" than previously reported. In my opinion greater precision, both in terms of acceptable outcomes and how they compared to previous research would strengthen the manuscript.

9. The discussion largely focused around the Bland-Altman results, which I believe is appropriate, but it may also be interesting to explicitly comment on the ICC and CCC outcomes. These appear to indicate excellent reliability, whereas the Bland-Altman analysis suggests important disagreement and higher and lower concentrations. It would be useful to explicitly explain the relevance of this finding, including the unique, but complementary information that both analyses provide. Currently, this is implied, rather than explicitly explained.

10. Discussion Paragraph 3: Prediction intervals and limits of agreement assess different parameters and as such should not be considered interchangeable. In the last line of this paragraphs it is suggested that the range measured in the previous studies was more limited than in this one, potentially explaining the smaller observed variability. However as this study also reports results

within a similar concentration range, it may be useful to compare agreement metrics within the most similar and overlapping concentration range.

11. What are the most likely analytical or methodological explanations for the greater bias observed at the higher and lower lactate concentrations? A brief discussion of potential mechanisms, along with consideration of how they may be overcome, would be very interesting and may act as a useful guide to planning future studies and developing more precise mobile devices in the future.

12. Practical recommendations: I appreciate the balance and pragmatism displayed here. I agree that many of these mobile devices can have utility, but it does depend on context and I believe you captured this well in your main take-home points.

Reviewer's Responses to Questions

Experimental quality

Does each figure have the proper controls?

If 'No', please indicate reasons in Comments for Author box below.

Reviewer #1:

- Yes

Reviewer #2:

- Yes

Were the data analyzed using appropriate statistical tests?

If 'No', please indicate reasons in Comments for Author box below.

Reviewer #1:

- Yes

Reviewer #2:

- Yes

Reproducibility

Were experiments performed using adequate number of biological replicates?

If 'No', please indicate reasons in Comments for Author box below.

Reviewer #1:

- Yes

Reviewer #2:

- Yes

Does the methods section provide sufficient detail to permit reproducibility?

If 'No', please indicate reasons in Comments for Author box below.

Reviewer #1:

- Yes

Reviewer #2:

- No

Completeness

Are the manuscript's conclusions supported by the data?

If 'No', please indicate reasons in Comments for Author box below.

Reviewer #1:

- Yes

Reviewer #2:

- Yes

Scholarship

Do the authors cite and discuss the merits of data that would argue for and against their conclusion?

If 'No', please indicate reasons in Comments for Author box below.

Reviewer #1:

- Yes

Reviewer #2:

- Yes

Does the manuscript title & abstract accurately reflect the contents of the manuscript, without hyperbole?

If 'No', please indicate reasons in Comments for Author box below.

Reviewer #1:

- Yes

Reviewer #2:

- Yes

First revision

Author response to reviewers' comments

In addition to the reviewers' comments, there is a concern about the conflict of interest. Transparency is important in conflicts of interest, whether real or perceived. In this case, the conflict of interest requires more prominent and critical acknowledgement throughout the paper. The Lactate Express was provided by the manufacturer's distributor in exchange for a symposium presentation by one of the authors, Dr. Renner. While this is disclosed at the end of the manuscript, its implications for the interpretation of results are not adequately discussed in the body of the manuscript. The overall framing of the device as suitable for threshold monitoring (despite having wider limits of agreement than comparable devices in the literature) should be presented with appropriate caution given this potential source of bias. Independent replication with commercially purchased devices would strengthen the evidence, and this lack of independent replication should be pointed out to readers. These points should be explicitly discussed in a "Limitations" section in the manuscript.

Furthermore, the COI statement will need modification. The affiliations of two authors with iq-move Praxis Fraunberger and one author with Schwarz Corporate Solutions KG, Health & Performance need descriptions, to alleviate perceived conflicts of interest.

Dear Editor,

thank you for this important comment. We agree that transparency regarding actual and perceived conflicts of interest is essential and that these issues should be acknowledged more explicitly in the manuscript.

We would like to clarify that we consider the study to have been conducted and interpreted independently. Neither the manufacturer, the distributor, nor any of their employees had any role in the study design, data collection, data analysis, interpretation of the findings, manuscript preparation, or the decision to submit the manuscript. They also did not review the manuscript prior to submission.

The Lactate Express device was provided by the distributor in connection with a symposium presentation by Dr. Peter Renner. Dr. Renner is a medical doctor at the practice where the study was conducted. Benedikt Meixner helps with data collection in the lab. The validation was partly motivated by practical interest in the usability of the device in field-based testing. However, this relationship did not influence the conduct or interpretation of the study. Our analyses identified clear range-dependent disagreement relative to the reference method, and we have revised the manuscript to reflect these findings with greater caution.

We nevertheless agree that the provision of the device by the distributor may represent a perceived conflict of interest and should therefore be discussed more prominently in the manuscript body. In response to this comment, we have added this point explicitly to the Limitations section and now state that the findings should be interpreted with appropriate caution.

In addition, we have revised the conflict-of-interest statement to clarify the affiliations of the authors connected to iq-move Praxis Fraunberger and Schwarz Corporate Solutions KG, Health & Performance. Dr. Silvia Achtzehn contributed her expertise in laboratory devices and diagnostics as a co-author, and these affiliations did not influence the collection, analysis, or interpretation of the data. Importantly, she joined Schwarz Corporate Solutions KG (Schwarz Group) after the completion of the data collection.

Furthermore, none of the authors received financial compensation related to this study, nor do any of the authors hold stocks, shares, or other financial interests in the manufacturer or distributor of the device. We appreciate this suggestion and agree that these revisions improve the transparency and balance of the manuscript.

Comments from the Reviewers:

Reviewer 1: This study addresses a practically relevant question regarding the analytical validity of a novel handheld blood lactate analyzer (Lactate Express) relative to an established point-of-care reference device (EKF Biosen C-Line). The stratified Bland-Altman approach across three physiologically meaningful concentration ranges is a clear strength of the manuscript and the inclusion of a threshold-based practical analysis adds meaningful applied value. However, there are several methodological, interpretive and presentational concerns that should be addressed before the manuscript is suitable for publication. The quality of the written English is generally acceptable, though a number of minor grammatical and typographical issues are noted below.

Abstract:

* The abstract is generally well written and conveys the key findings.

* However, a full stop is missing at the end of the Main results section ("–1.03 mmol·L⁻¹Differences"); this should be corrected.

Thank you for these comments. We now added the full stop.

Introduction:

The introduction is concise and adequately motivates the study. The relevant literature on handheld lactate device validation is cited and the aim is clearly stated. A few points should be addressed:

* There is a full stop immediately prior to the reference in the first sentence. This should be removed.

Revised as suggested.

* A reference would be helpful to support the statement "While stationary laboratory-grade devices are widely regarded as the gold standard".

* The introduction frames the EKF Biosen C-Line as an "established stationary reference" device. However, little detail is provided regarding this device. It is critical to the validity of the manuscript that the EKF Biosen C-Line is indeed the gold standard reference, so I would ask that the authors justify this selection.

The EKF devices are the most common laboratory devices in Germany and are often used as reference benchtop devices. We now rephrased these sentences to reflect the overall use of a laboratory device as reference device with references and also include references that used the EKF devices as reference device.

* The final paragraph of the introduction effectively states the aim twice (once in general terms and again in specific terms). Consider condensing these into a single, clearly structured aim statement.

We now condensed these sentences as suggested.

Methods:

* The participant group was heavily biased towards males (27 men and 8 women). This is an ongoing problem in the physiology literature, so the authors should provide a justification for this bias or acknowledge it as a limitation. Secondly, given that physiological differences between sexes could plausibly influence lactate kinetics and the distribution of concentration ranges in the dataset, the authors should either acknowledge this as a limitation or confirm that such an imbalance did not affect the results.

We agree that sex-related physiological differences may influence lactate kinetics and the distribution of observed lactate concentrations. However, the primary aim of the present study was the analytical and practical validation of the device across the sampled concentration range rather than the characterization of sex-specific lactate responses. As the comparison relied on paired measurements obtained from the same samples, the main outcomes focused on agreement, bias, and precision between the analyzers. For this reason, we consider it unlikely that the sex imbalance substantially influenced the central method-comparison results. Nevertheless, because the sample was not balanced for sex, potential sex-specific differences in device performance cannot be ruled out. We therefore acknowledge this aspect as a limitation in the revised manuscript.

* Participants were "recruited from routine clinical practice according to their health status and the availability of the laboratory assistant." This description raises questions about selection bias and the representativeness of the sample. Were any health-related exclusion criteria applied? Were participants screened for conditions that might affect lactate metabolism? This should be clarified.

In response to this and the other reviewer as well, we now state that the participants represent a convenience sample. If the participant's health history was known and indicated possible health issues, we would have excluded them from this study. Furthermore, if in the other diagnostic procedures not related to this study but performed during the visit any indication would be found that might influence lactate metabolism, we would exclude the collected data. This however was not the case. We now clarified that in the manuscript.

* Table 1 might be improved by splitting by sex.

We are absolutely prepared to do so if deemed necessary, but as the aim of this methods article to validate the handheld device against a bench top analyzer, we prefer not to split the data into - as is usually the case in other validation studies for handheld lactate analyzers.

Instead, in agreement with the other comment, we now provide the participant data of those included in the threshold analysis.

* Testing protocol: The city name "Leipzig" in the Cycclus2 ergometer description appears to be a typographical error; it should read "Leipzig."

Thank you for noticing this, we now corrected this typo.

* Testing protocol: There is another rogue full stop prior to a reference (line 55). Please check throughout.

Thank you for noticing this. We now checked this throughout the manuscript.

* The Lactate Express was calibrated using a "manufacturer-provided reference solution, yielding satisfactory calibration results." What were the calibration target values? How was "satisfactory" defined? Please provide the specific calibration outcomes or at minimum state the acceptable tolerance criteria used.

We now include this information.

* The lactate threshold analysis was restricted to participants with ≥ 6 complete measurements and no missing values, resulting in only 8 runners and 8 cyclists being analysed. This is a notably small subset of the full cohort and limits the generalisability of the threshold comparison. Could the authors please report the characteristics (e.g., age, fitness level, lactate range) of this subset relative to the full sample, clearly state the sex of these 8 runners and 8 cyclists and more explicitly acknowledge the limited statistical power of this analysis.

We now provide this data and acknowledge it in our discussion¹.

Statistical analysis:

* The analytical approach (ICC, CCC, MAE, and stratified Bland-Altman) is well chosen and clearly described. The use of a two-way random-effects model for absolute agreement is appropriate given the study design.

Thank you for this comment.

* The sample sizes across concentration ranges are considerably unequal: $n=36$ ($<1 \text{ mmol}\cdot\text{L}^{-1}$), $n=125$ ($1-4 \text{ mmol}\cdot\text{L}^{-1}$) and $n=70$ ($>4 \text{ mmol}\cdot\text{L}^{-1}$). This imbalance is not acknowledged or discussed and may be particularly relevant in the $<1 \text{ mmol}\cdot\text{L}^{-1}$ range (where the smaller sample may reduce the reliability of the LoA estimate). The authors might discuss the implications of these unequal group sizes for the interpretation of range-specific agreement.

Thank you for this important point. We agree that unequal data availability across concentration ranges affects the precision of the range-specific estimates and should be acknowledged when interpreting the Bland-Altman results. In our dataset, the number of paired samples was highest in the $1-4 \text{ mmol}\cdot\text{L}^{-1}$ range ($n = 125$) and lower at the extremes ($<1 \text{ mmol}\cdot\text{L}^{-1}$: $n = 36$; $>4 \text{ mmol}\cdot\text{L}^{-1}$: $n = 70$), which increases uncertainty particularly for the low- and high-concentration range estimates. However, the observed pattern was not consistent with a sample-size artefact alone, as bias was close to zero in the $1-4 \text{ mmol}\cdot\text{L}^{-1}$ range but clearly shifted at the lower and higher ends of the concentration range. We have now acknowledged this issue in the manuscript and provide additional Bland-Altman summary information in the supplementary material.

Results:

* The results are concise and the figures are clear. The Bland-Altman plots are well presented, and the stratified analysis effectively illustrates concentration-dependent behaviour.

Thank you for this comment.

* Table 1 is described in the text as containing "descriptive results," but the table as presented contains only participant characteristics. The descriptive lactate data (medians, ranges etc.) are in Table 2. Please ensure the in-text references to tables are consistent with the tables as labelled.

We now ensured consistent labelling in our text throughout the manuscript.

* Table 3 has 2 blank lines at the bottom - I'd suggest this should be cleaned up.

Revised as suggested.

* The text mentions Figures 6 and 7 for the threshold comparisons ("displayed in figure 6 for running and figure 7 for cycling"), but only Figure 4 is present in the manuscript. This appears to be a figure numbering inconsistency that should be resolved.

Indeed, we originally included all figures separately but present them more condensed as panels in the figures, reducing figure numbers. We now adjusted this accordingly.

* The image quality of some of the figures is really quite poor. This should be enhanced.

We agree that the quality in the pdf is poor. However, when we downloaded the files per the link given on the upper right of each page, we were able to download our own files in good quality. We were unable to find a solution to the problem of poor quality in the pdf, but will keep extra attention to good quality during production.

Discussion:

The discussion engages well with the prior literature and the concentration-range framing provides a useful structure. However, several points may require attention:

* It seems as though there should be a new paragraph and line break at line 9/10?

Revised as suggested.

* Lactate Express is sometimes in italics, sometimes not. Please correct this inconsistency.

We now use italics for the device throughout the manuscript.

* There should be a space after the full stop on line 22. Please correct and check throughout.

We now checked this throughout the manuscript.

* The practical recommendations offered in the discussion (e.g., "prioritize consistency in the choice of analyzer", avoid mixing handheld and laboratory data") are sensible and appropriate. However, this paragraph refers to Figures 6 and 7, which don't appear to exist.

As stated in a previous comment, we explain why this happened and corrected this.

* The limitations paragraph is brief and would benefit from expansion. Please add the limitations I mention earlier in my review.

Revised as suggested by the editor and the reviewers.

In summary, this is a useful and practically motivated validation study that addresses a clear gap in the literature. The stratified Bland-Altman approach is particularly commendable. Addressing the points above would substantially strengthen the manuscript.

Reviewer 2: The authors have conducted a validation study comparing a handheld lactate analyzer with a laboratory reference device. They reported good relative agreement, but concentration

dependant bias in terms of absolute agreement, with greater discrepancies between analysis approaches apparent at both higher and lower lactate concentrations.

Overall, the article approaches an interesting and relevant topic that aligns well with the methods and techniques article description. Hand-held devices such as the lactate analyser under investigation here have large utility in practice, or for field-research, but validation against reference devices is important. In my opinion, the authors have conducted an interesting and useful study, and their conclusions are balanced and directly informed by their results. There were some aspects of the manuscript that I felt would benefit from further elaboration or clarification, however. Several points were implied rather than directly stated, which can lead to misunderstanding and impact accurate interpretation. I have detailed the points that I believe warrant additional attention below.

Specific Comments:

1. The article largely focuses on the use of the Lactate Express in an exercise physiology context, however as identified by the authors in their opening sentence, lactate assessment has utility in wide-ranging contexts, including, but not limited to exercise physiology. Considering that this is a general Biology journal, I believe that it should not be assumed that all readers will be familiar with concepts such as threshold identification and their implications for program design and that some elaboration on these concepts, and justification for the design decisions made (e.g., the relevance of the 2 and 4 mmol thresholds) as well as consideration of the other contexts in which these mobile lactate analysers may be used would be useful.

Thank you for this comment. We now provide more information on this in our discussion. However, the Lactate Express is explicitly marketed towards sports and exercise applications.

2. Methods section: There were some parts of the first two paragraphs that I was unclear on. Were the participants in this study purposefully recruited or was this a convenience sample conducted on individuals who were undertaking incremental tests for other purposes? The statement on lines 6 - 7 ("performed either for health assessment or performance diagnostics") implies the latter, but it is important to be explicit on this point. Were any eligibility criteria applied to participant selection, e.g., in terms of age, health or training status? The statement on line 25 - 26 implies that there were ("recruited according to their health status") but the details are unclear. Greater precision, clarity and accuracy is required throughout the methods to ensure the reader fully understands the details of the study.

Thank you for this comment. It was indeed a convenience sample of participants undergoing exercise testing. We now added the information that participants with conditions that might influence lactate metabolism or kinetics would have been excluded if this was either known before by previous visits or diagnostics or became evident through other diagnostic procedures during the visit. This was however not the case.

3. Methods - Capillary Blood Lactate measurements: It is stated that a single drop of capillary blood was obtained from the left earlobe, enabling simultaneous measurements using both devices. Considering the large difference in required volume (0.8 vs 20 μ L) how was volume separation managed in practice? Did you have any issues in ensuring sufficient volume for each analysis and were there any issues in time delay between sampling and measurement? The alternation of assessment order is likely to at least partly overcome any potential issues, however greater clarity on how the design was applied in practice would be useful.

We aimed to produce as few drops as possible and either used the first or last drop for both capillaries and stripes of both devices. We now clarify this in our manuscript. Participants in which it was evident to be challenging to produce enough blood for both measurements in a timely manner would have been excluded.

4. Methods - Lactate threshold calculation section: If I have understood correctly the threshold analysis was conducted on 16 individuals, presumably those who were undergoing performance based tests and for whom a larger number of data-points were available. This is important

information that warrants further consider, e.g., in the abstract it is implied that the threshold data is available for the full dataset of 35 participants. Conducting this analysis only on a substantially reduced participant subset has important statistical implications that should be considered.

We now clarify this in our abstract. We further now also explicitly state this in our limitations.

5. Following on from the above point, I believe that data availability for the different assessments made is an important consideration that warrants some further attention. How much of an influence does sample size have on these analyses? It is interesting that the concentration ranges that showed the greater biases also had substantially fewer data-points for evaluation. Is it possible that data availability may have impacted these results?

We thank the reviewer for this important point. We agree that unequal data availability across concentration ranges affects the precision of the range-specific estimates and should be acknowledged when interpreting the Bland-Altman results. In our dataset, the number of paired samples was highest in the 1-4 mmol·L⁻¹ range (n = 125) and lower at the extremes (<1 mmol·L⁻¹: n = 36; >4 mmol·L⁻¹: n = 70), which increases uncertainty particularly for the low- and high-concentration range estimates. However, the observed pattern was not consistent with a sample-size artefact alone. Bias was close to zero in the 1-4 mmol·L⁻¹ range (-0.05 mmol·L⁻¹; 95% CI -0.11 to 0.01), but clearly positive below 1 mmol·L⁻¹ (+0.29 mmol·L⁻¹; 95% CI 0.23 to 0.35) and clearly negative above 4 mmol·L⁻¹ (-1.03 mmol·L⁻¹; 95% CI -1.31 to -0.75). We therefore interpret lower data availability as contributing to uncertainty in the range-specific estimates, but not as a sufficient explanation for the concentration-dependent disagreement observed between devices. We now include a table of Bland-Altman-analysis including SD of bias and LoA in our supplementary files.

6. The dataset includes 231 paired lactate measurements obtained from 35 participants, meaning that multiple observations originate from the same individuals and are likely to be highly correlated. From what I understand, the statistical analyses treat these observations as independent, which may potentially impact the agreement metrics, particularly the ICC and Bland-Altman limits of agreement. Was dependency considered within the analysis? If not, it would be useful to consider the potential influence of this on the main outcomes of interest.

We thank the reviewer for this important comment. We agree that the dataset contains repeated paired measurements within the same participants and that these observations are therefore not fully independent. In the original analysis, this dependency structure was not modelled explicitly. We acknowledge that within-subject clustering may affect the precision of some agreement estimates, particularly confidence intervals and limits of agreement, and may also influence correlation-based indices such as the ICC. However, because the central aim of the study was to characterize practical agreement patterns between devices across the observed concentration range, we believe the main pattern of the findings remains informative. We have now acknowledged this issue explicitly as a limitation in the manuscript. In addition, we clarified that the concentration categories were chosen pragmatically to reflect low, moderate, and high lactate ranges relevant to practical interpretation, rather than being derived from formal statistical cut-points.

7. What was the purpose of the MAE assessment, considering that the Bland-Altman provides similar information? I recognize that these analyses can provide related and complementary information, and it would be useful to be more explicit on their unique contributions to the main question. It would also be useful to calculate confidence intervals around the MAE to give an indication of certainty in the estimate and to elaborate on the implications of this analysis in the results and discussion.

Thank you for this comment. We now clarify more explicitly that Bland-Altman analysis was used to assess agreement structure, including systematic bias, limits of agreement, and concentration-dependent disagreement, whereas the MAE was included as a complementary measure of the typical absolute between-device deviation expressed directly in mmol·L⁻¹, irrespective of direction.

Following the reviewer's suggestion, we also report a bootstrap 95% confidence interval for the MAE and elaborate on its practical interpretation in the Results and Discussion.

8. Were any predefined acceptance criteria for the various analyses used? Based on the discussion, it appears not, and at several points somewhat vague phrases are used, e.g., "substantial limits of agreement" and LoA which were "somewhat wider" than previously reported. In my opinion greater precision, both in terms of acceptable outcomes and how they compared to previous research would strengthen the manuscript.

No formal a priori acceptance criteria were defined for the agreement analyses, as we are not aware of universally accepted decision thresholds for handheld-versus-bench-top lactate analyzer agreement in this specific exercise-testing context. We therefore interpreted the results in relation to the magnitude and direction of the observed disagreement, previously published validation data, and the practical implications for threshold-based exercise testing. We have also revised the manuscript to reduce qualitative wording and to compare our findings more explicitly and quantitatively with previous work.

9. The discussion largely focused around the Bland-Altman results, which I believe is appropriate, but it may also be interesting to explicitly comment on the ICC and CCC outcomes. These appear to indicate excellent reliability, whereas the Bland-Altman analysis suggests important disagreement and higher and lower concentrations. It would be useful to explicitly explain the relevance of this finding, including the unique, but complementary information that both analyses provide. Currently, this is implied, rather than explicitly explained.

We thank the reviewer for this important comment. We agree that the complementary interpretation of ICC/CCC and Bland-Altman analysis should be stated more explicitly. In the revised manuscript, we now clarify that ICC and CCC reflect overall consistency and concordance across the full dataset and therefore indicate how well the analyzer preserves the ranking and general correspondence of lactate values. In contrast, Bland-Altman analysis assesses absolute agreement, including systematic bias, limits of agreement, and the concentration-dependent pattern of disagreement. Thus, high ICC/CCC values do not imply interchangeability when devices track each other consistently but differ meaningfully in absolute terms, particularly at the lower and higher ends of the lactate range. We have added this explanation to the Discussion.

10. Discussion Paragraph 3: Prediction intervals and limits of agreement assess different parameters and as such should not be considered interchangeable. In the last line of this paragraphs it is suggested that the range measured in the previous studies was more limited than in this one, potentially explaining the smaller observed variability. However as this study also reports results within a similar concentration range, it may be useful to compare agreement metrics within the most similar and overlapping concentration range.

We thank the reviewer for this important comment. We agree that prediction intervals and Bland-Altman limits of agreement assess different statistical properties and should not be treated as interchangeable. We have revised the Discussion accordingly. We also agree that comparison with previous studies is most informative within the most similar concentration range. Therefore, we now explicitly note that Mentzoni et al. assessed a more restricted lactate range (0.88-4.89 mmol·L⁻¹), and we contrast this with our own 1-4 mmol·L⁻¹ results, where bias was near zero (-0.05 mmol·L⁻¹) and limits of agreement were considerably narrower than in the full dataset. This supports the interpretation that part of the larger overall variability in the present study is attributable to the inclusion of lower and higher lactate concentrations, where disagreement between devices became more pronounced.

11. What are the most likely analytical or methodological explanations for the greater bias observed at the higher and lower lactate concentrations? A brief discussion of potential mechanisms, along with consideration of how they may be overcome, would be very interesting and may act as a useful guide to planning future studies and developing more precise mobile devices in the future.

We now added a speculation about potential mechanisms and future directions.

12. Practical recommendations: I appreciate the balance and pragmatism displayed here. I agree that many of these mobile devices can have utility, but it does depend on context and I believe you captured this well in your main take-home points.

Thank you for this comment.

Second decision letter

MS ID#: bio.062543R1

MS Title: Validation of a Novel Handheld Lactate Analyzer Reveals Concentration-Dependent Bias Compared with a Laboratory Reference Device

Authors: Benedikt Johannes Meixner, Manuel Matzka, Peter Renner, Silvia Achtzehn, Billy Sperlich

I have now reached a decision on the above manuscript.

The reviewer reports are shown at the bottom of this email.

As you will see, the reviewers gave favourable reports, but reviewer 2 had a few additional comments that should be easy to address. I hope that you will be able to carry these out, because we would like to be able to accept your paper.

At this stage, we also ask you to ensure your manuscript complies with our formatting guidelines - please see our manuscript preparation guidelines for details. Provided you are able to fully address the referees' comments, we are positive about publication of your paper (we accept over 95% of revision submissions) and therefore hope you won't mind any extra work involved in reformatting your manuscript at this point.

Please upload both a 'clean' version of your Word file, along with a highlighted version clearly showing where you have made changes in the revised manuscript. Please avoid using 'Track changes' in Word files as these are lost in PDF conversion.

I should be grateful if you would also provide a point-by-point response detailing how you have dealt with the points raised by the reviewers in the 'Response to Reviewers' box. Please attend to all of the reviewers' comments. If you do not agree with any of their criticisms or suggestions please explain clearly why this is so.

Reviewer 1

Comments for the authors

Dear Authors,

Thank you for your thorough and thoughtful response to my review comments. I am satisfied that you have addressed all the points I raised appropriately. The revisions you have made have strengthened the manuscript considerably, particularly with regard to methodological transparency, acknowledgment of limitations, and the handling of potential conflicts of interest.

I am pleased to recommend your manuscript for acceptance.

Best wishes,

Reviewer 2

Comments for the authors

I thank the authors for their detailed response to my original suggestions. In my opinion the article, and particularly the discussion, is much improved. Below are three additional minor suggestions that the authors may wish to consider for their final article.

1. Abstract significance - line reading "The Lactate Express demonstrates good relative agreement but only limited absolute agreement within the 1-4mmol/L range". Is this right? My understanding is that the reports indicated very close absolute agreement at the 1-4 range, but that substantial concentration dependent bias was observed at the higher and lower ranges.

2. Discussion paragraph 3: I believe the additions made to the end of this paragraph are important, and it does seem likely that the more limited range investigated in the study by Mentzoni et al. largely explains their apparently tighter ranges, particularly given the results from the current study indicating that absolute bias occurs only at higher and lower concentrations. This point could potentially be stated more explicitly, as the restricted range likely excludes the regions where variability and bias are greatest.

In contrast, the addition to the middle paragraph (beginning "however, prediction intervals.....) is unclear and conceptually inconsistent. It is correct to state that prediction intervals and Bland-Altman LoA describe different statistical quantities and are not directly comparable, but the text continues to directly compare them. A minor re-phrasing may be useful to decouple these issues and avoid potential confusion, e.g., something along the lines of the following:

"In a study comparing four handheld lactate devices (Lactate Plus, Lactate Pro2, Lactate Scout 4 and TaiDoc TD-4289) to an EKF Biosen device (Mentzoni et al. 2024), the handheld systems appeared to demonstrate narrower intervals than those observed in the present study. It should be noted that prediction intervals and Bland-Altman limits of agreement describe different statistical quantities and are not directly comparable. As such, differences in interval width should be interpreted with caution and not taken as evidence of superior agreement. In addition, the lactate concentration range assessed in Mentzoni et al. (0.88-4.89 mmol·L⁻¹) was relatively restricted. Given the present findings of concentration-dependent bias, with greater discrepancies observed at lower and higher concentrations, this narrower ranges may reduce observed variability, as it does not capture the greater discrepancies that may occur across a wider physiological range".

3. Discussion - third last paragraph: Please provide citations to the previous studies that are referred to in the 2nd line.

Reviewer's Responses to Questions

Experimental quality

Does each figure have the proper controls?

If 'No', please indicate reasons in Comments for Author box below.

Reviewer #1:

- Yes

Reviewer #2:

- Yes

Were the data analyzed using appropriate statistical tests?

If 'No', please indicate reasons in Comments for Author box below.

Reviewer #1:

- Yes

Reviewer #2:

- Yes

Reproducibility

Were experiments performed using adequate number of biological replicates?

If 'No', please indicate reasons in Comments for Author box below.

Reviewer #1:

- Yes

Reviewer #2:

- Yes

Does the methods section provide sufficient detail to permit reproducibility?

If 'No', please indicate reasons in Comments for Author box below.

Reviewer #1:

- Yes

Reviewer #2:

- Yes

Completeness

Are the manuscript's conclusions supported by the data?

If 'No', please indicate reasons in Comments for Author box below.

Reviewer #1:

- Yes

Reviewer #2:

- Yes

Scholarship

Do the authors cite and discuss the merits of data that would argue for and against their conclusion?

If 'No', please indicate reasons in Comments for Author box below.

Reviewer #1:

- Yes

Reviewer #2:

- Yes

Does the manuscript title & abstract accurately reflect the contents of the manuscript, without hyperbole?

If 'No', please indicate reasons in Comments for Author box below.

Reviewer #1:

- Yes

Reviewer #2:

- Yes

Second revision

Author response to reviewers' comments

Reviewer 1: Dear Authors,

Thank you for your thorough and thoughtful response to my review comments. I am satisfied that you have addressed all the points I raised appropriately. The revisions you have made have strengthened the manuscript considerably, particularly with regard to methodological transparency, acknowledgment of limitations, and the handling of potential conflicts of interest.

I am pleased to recommend your manuscript for acceptance.

Best wishes,

Reviewer 1

Thank you for this comment and, in total, the whole review process. We appreciate the timeliness and constructive feedback!

Reviewer 2: I thank the authors for their detailed response to my original suggestions. In my opinion the article, and particularly the discussion, is much improved. Below are three additional minor suggestions that the authors may wish to consider for their final article.

1. Abstract significance - line reading "The Lactate Express demonstrates good relative agreement but only limited absolute agreement within the 1-4mmol/L range". Is this right? My understanding is that the reports indicated very close absolute agreement at the 1-4 range, but that substantial concentration dependent bias was observed at the higher and lower ranges.

Indeed. We now corrected this statement.

2. Discussion paragraph 3: I believe the additions made to the end of this paragraph are important, and it does seem likely that the more limited range investigated in the study by Mentzoni et al. largely explains their apparently tighter ranges, particularly given the results from the current study indicating that absolute bias occurs only at higher and lower concentrations. This point could potentially be stated more explicitly, as the restricted range likely excludes the regions where variability and bias are greatest.

In contrast, the addition to the middle paragraph (beginning "however, prediction intervals.....") is unclear and conceptually inconsistent. It is correct to state that prediction intervals and Bland-Altman LoA describe different statistical quantities and are not directly comparable, but the text continues to directly compare them. A minor re-phrasing may be useful to decouple these issues and avoid potential confusion, e.g., something along the lines of the following:

"In a study comparing four handheld lactate devices (Lactate Plus, Lactate Pro2, Lactate Scout 4 and TaiDoc TD-4289) to an EKF Biosen device (Mentzoni et al. 2024), the handheld systems appeared to demonstrate narrower intervals than those observed in the present study. It should be noted that prediction intervals and Bland-Altman limits of agreement describe different statistical quantities and are not directly comparable. As such, differences in interval width should be interpreted with caution and not taken as evidence of superior agreement. In addition, the lactate concentration range assessed in Mentzoni et al. (0.88-4.89 mmol·L⁻¹) was relatively restricted. Given the present findings of concentration-dependent bias, with greater discrepancies observed at lower and higher concentrations, this narrower ranges may reduce observed variability, as it does not capture the greater discrepancies that may occur across a wider physiological range".

Thank you for this suggestion, we revised it accordingly.

3. Discussion - third last paragraph: Please provide citations to the previous studies that are referred to in the 2nd line.

Thank you for this comment, we now added references for this.

Furthermore, we also want to thank you for the constructive feedback and the fast review process.

Third decision letter

MS ID#: bio.062543R2

MS Title: Validation of a Novel Handheld Lactate Analyzer Reveals Concentration-Dependent Bias Compared with a Laboratory Reference Device

Authors: Benedikt Johannes Meixner, Manuel Matzka, Peter Renner, Silvia Achtzehn, Billy Sperlich

I am happy to tell you that your manuscript has been accepted for publication in Biology Open, pending our standard publication integrity checks. It was accepted on 24th March 2026. We appreciate the positive feedback about our review process.